# Analysis of Vibration and Acoustic Signals for Noncontact Measurement of Engine Rotation Speed

**DOI:** 10.3390/s20030683

**Published:** 2020-01-26

**Authors:** Xuansheng Shan, Lu Tang, He Wen, Radek Martinek, Janusz Smulko

**Affiliations:** 1College of Electrical and Information Engineering, Hunan University, Changsha 410082, China; shanxs006@hnu.edu.cn (X.S.); tangl@126.com (L.T.); 2Department of Cybernetics and Biomedical Engineering, Technical University of Ostrava, Office EA339 17. Listopadu 2172/15, 708 00 Ostrava-Poruba, Czech Republic; radek.martinek@vsb.cz; 3Department of Metrology and Optoelectronics, Gdańsk University of Technology, ul. G. Narutowicza 11/12, 80-233 Gdańsk, Poland; janusz.smulko@pg.edu.pl

**Keywords:** engine rotation speed, cross-correlation, vibration, acoustic, noncontact measurement, Fourier transform

## Abstract

The non-contact measurement of engine speed can be realized by analyzing engine vibration frequency. However, the vibration signal is distorted by harmonics and noise in the measurement. This paper presents a novel method for the measurement of engine rotation speed by using the cross-correlation of vibration and acoustic signals. This method can enhance the same frequency components in engine vibration and acoustic signal. After cross-correlation processing, the energy centrobaric correction method is applied to estimate the accurate frequency of the engine’s vibration. This method can be implemented with a low-cost embedded system estimating the cross-correlation. Test results showed that this method outperformed the traditional vibration-based measurement method.

## 1. Introduction

The rapid development of the automobile industry has put forward higher requirements for efficient automobile inspection methods. The engine rotation speed is a key parameter for evaluating an automobile’s condition. For example, when testing an automobile’s emissions, it is necessary to keep the engine’s rotation speed within a certain range. Thus, the fast and accurate measurement of the engine rotation speed is of great importance during the automobile inspection. 

Different methods are used for the measurement of engine rotation speed, which can be divided into two types (i.e., the contact-type and noncontact-type). The contact-type measurement is usually applied to unarmed engines or some engines with pre-installed sensors [1]. For example, in [2] the authors propose a direct measurement system based on magnetoelectric transducers, which shows good performance in the accuracy and reliability of rotation speed measurement. This method requires the installation of a sensor into the internal structure of the engine. However, during automobile inspection, it is impossible to disassemble the engine and install sensors inside. So, the contact-type methods are impractical for automobile inspection. 

In automobile inspection, noncontact-type measurement plays an important role [3,4,5,6]. According to the structure and working principle of an engine, the traditional noncontact-type measurement is based on the inherent relationship between the frequency of vibration and rotation speed. The advantage of noncontact-type measurement is the simple operation process, because the signal can be easily acquired by arranging vibration or acoustic sensors [7,8]. However, there are some limitations of noncontact-type measurements. In rotation speed measurement, embedded hardware devices are usually used, which means that the computing power of the hardware is limited. Therefore, a method with too great a computational cost is impractical. Among the mainstream frequency estimation methods, time-domain (parametric) methods [9,10,11] require computationally intensive algorithms to achieve better performance [12], while frequency-domain (non-parametric) methods [13,14,15,16,17] provide an accurate estimation of frequency with relatively low computational burden and better anti-noise performance. Therefore, frequency-domain methods are more suitable for rotation speed measurement. However, the signals are not only affected by noise, harmonic interference is also a cause of large deviation in fundamental frequency estimation. Taking vibration-based methods as an example, during the idling stage and accelerating stage, when the energy of interference is stronger than the fundamental frequency component, vibration-based methods cannot provide reliable results [18]. The same problem also occurs in methods based on acoustic signals. Thus, the reliability of the measurement result cannot be ensured if using only the vibration signal or the acoustic signal without eliminating interference.

The characteristic of vibration and acoustic signals was studied [19], and the EMD (Empirical Mode Decomposition) method was applied to extract the common feature in vibration and acoustic signals; however, a joint analysis was not considered. Correlation methods are applied in various fields of mechanical engineering and the cross-correlation function has been applied to compare the surfaces of rotary elements [20]. Cross-correlation is also used to measure the similarity of two sets of signals, and the correlation sequence contains the characteristics of two sets of signals [21]. Taking into account the intrinsic connection between the vibration and acoustic signals, this paper proposes a noncontact method of measuring engine rotation speed based on cross-correlation. The rest of the paper is organized in three parts. The architecture of the proposed method is introduced in the first part, including the measurement principle, the cross-correlation of vibration and acoustic signals, and estimation of fundamental frequency. Then, an embedded hardware design is proposed, and the test results are compared with a vibration-based method. Conclusions are drawn in the last part.

## 2. Method 

### 2.1. Measurement Principle

In general, a normally functioning engine generates a regular vibration which is closely related to the rotation speed [22]. In the meantime, the fuel combustion in the engine cylinders causes a deterministic tone in the engine noise, and the frequency of the deterministic tone is equal to the fundamental frequency of vibration. The mathematical relationship between rotation speed and fundamental frequency is:(1)$$RPM=60\left(T/2\right)\left(f_{0}/C\right),$$
where *RPM* represents revolutions per minute, *T* is the number of strokes, *f*_0_ is the fundamental frequency of vibration and acoustic signals, and *C* is the number of cylinders [23,24]. From Equation (1), it is clear that the ratio between *RPM* and *f*_0_ is constant when *T* and *C* are set; that is, for a four-stroke diesel engine with four cylinders, the rotation speed can be computed through:(2)$$RPM=30f_{0}.$$

The frame of the measurement system was designed as shown in Figure 1. The vibration and acoustic signals are picked up by sensors and a low-pass filter is applied to restrain the high-frequency interference. After that, cross-correlation is applied to the filtered vibration and acoustic signals. Then, a frequency estimation algorithm based on fast Fourier transform is applied to calculate the frequency of the obtained sequence and to get the rotation speed through Equation (1).

### 2.2. Cross-Correlation of Vibration and Acoustic Signals

In addition to the fundamental frequency component, there are interference components existing in the vibration and acoustic signals which are generated by other parts of the vehicle. In order to restrain interferences and extract the fundamental frequency component of vibration and acoustic signals, a cross-correlation algorithm is applied.

In the process of sound and vibration propagation, the interferences in the two signals are usually not the same. Supposing that *v*(*t*) represents the vibration signal and *a*(*t*) represents the acoustic signal, the model of vibration and acoustic signals can be formulated as:(3)$$v\left(t\right)=A_{0}\cos\left(2\pi f_{0}t+\varphi_{0}\right)+\sum_{i=1}^{M}A_{vi}\cos\left(2\pi f_{vi}t+\varphi_{i}\right)+n_{v}\left(t\right),$$
(4)$$a\left(t\right)=A_{0}\cos\left(2\pi f_{0}t+\varphi_{0}\right)+\sum_{i=1}^{N}A_{ai}\cos\left(2\pi f_{ai}t+\varphi_{i}\right)+n_{a}\left(t\right),$$
where *f*_0_ is the common frequency of vibration and acoustic signals, which is closely related to the rotation speed; *f_vi_* and *f_ai_* represent the frequency of the interference signals of the vibration signal and acoustic signal respectively, *f_vi_* ≠ *f_ai_*. *n_v_*(*t*) and *n_a_*(*t*) are additive Gaussian white noise components of the vibration signal and acoustic signal, respectively. 

Considering that the model of vibration and acoustic signals is the combination of sinusoids with different frequencies and noise, the correlation sequence of vibration and acoustic signals can be rearranged as a linear superposition of several parts [25]. For simplicity, it is divided into three parts. The first part is the correlation of common frequency components. The second part is the correlation of different frequency components, and the third part is the correlation of noise components and frequency components, which are presented below:(5)$$r_{1}\left(\tau \right)=\underset{T\to \infty }{\lim}\int_{-\frac{T}{2}}^{\frac{T}{2}}A_{0}^{2}\cos\left(2\pi f_{0}t+\varphi_{0}\right)\cos\left(2\pi f_{0}\left(t-\tau \right)+\varphi_{0}\right)dt$$
(6)$$\begin{array}{cc} r_{2}\left(\tau \right) & =\underset{T\to \infty }{\lim}\int_{-\frac{T}{2}}^{\frac{T}{2}}A_{0}\cos\left(2\pi f_{0}t+\varphi_{0}\right)\sum_{i=1}^{N}A_{ai}\cos\left(2\pi f_{ai}\left(t-\tau \right)+\varphi_{i}\right)dt\\ & +\underset{T\to \infty }{\lim}\int_{-\frac{T}{2}}^{\frac{T}{2}}A_{0}\cos\left(2\pi f_{0}\left(t-\tau \right)+\varphi_{0}\right)\sum_{i=1}^{M}A_{vi}\cos\left(2\pi f_{vi}t+\varphi_{i}\right)dt \end{array}$$
(7)$$\begin{array}{ll} r_{3}\left(\tau \right) & =\underset{T\to \infty }{\lim}\int_{-\frac{T}{2}}^{\frac{T}{2}}n_{v}\left(t\right)\left(A_{0}\cos\left(2\pi f_{0}\left(t-\tau \right)+\varphi_{0}\right)+\sum_{i=1}^{N}A_{ai}\cos\left(2\pi f_{ai}\left(t-\tau \right)+\varphi_{i}\right)\right)dt\\ & +\underset{T\to \infty }{\lim}\int_{-\frac{T}{2}}^{\frac{T}{2}}n_{a}\left(t-\tau \right)\left(A_{0}\cos\left(2\pi f_{0}\left(t\right)+\varphi_{0}\right)+\sum_{i=1}^{M}A_{vi}\cos\left(2\pi f_{vi}t+\varphi_{i}\right)\right)dt\\ & +\underset{T\to \infty }{\lim}\int_{-\frac{T}{2}}^{\frac{T}{2}}n_{v}\left(t\right)n_{a}\left(t-\tau \right)dt \end{array}$$

The frequency domain characteristic of each part can be obtained by Fourier transform. According to the characteristics of cross-correlation and Fourier transform, the frequency spectrum of each part is expressed as below:(8)$$\begin{array}{ll} R_{1}\left(j\omega \right) & =F\left(r_{1}\right)\\ & =\left(A_{0}\pi \left(e^{j\varphi_{0}}\delta \left(\omega +2\pi f_{0}\right)+e^{-j\varphi_{0}}\delta \left(\omega -2\pi f_{0}\right)\right)\right){\left(A_{0}\pi \left(e^{j\varphi_{0}}\delta \left(\omega +2\pi f_{0}\right)+e^{-j\varphi_{0}}\delta \left(\omega -2\pi f_{0}\right)\right)\right)}^{*} \end{array}$$
(9)$$\begin{array}{l} R_{2}\left(j\omega \right)=F\left(r_{2}\right)\\ =\left(A_{0}\pi \left(e^{j\varphi_{0}}\delta \left(\omega +2\pi f_{0}\right)+e^{-j\varphi_{0}}\delta \left(\omega -2\pi f_{0}\right)\right)\right){\left(\sum_{i=1}^{M}A_{ai}\pi \left(e^{j\varphi_{i}}\delta \left(\omega +2\pi f_{ai}\right)+e^{-j\varphi_{i}}\delta \left(\omega -2\pi f_{ai}\right)\right)\right)}^{*}\\ +{\left(A_{0}\pi \left(e^{j\varphi_{0}}\delta \left(\omega +2\pi f_{0}\right)+e^{-j\varphi_{0}}\delta \left(\omega -2\pi f_{0}\right)\right)\right)}^{*}\left(\sum_{i=1}^{N}A_{vi}\pi \left(e^{j\varphi_{i}}\delta \left(\omega +2\pi f_{vi}\right)+e^{-j\varphi_{i}}\delta \left(\omega -2\pi f_{vi}\right)\right)\right) \end{array}$$
(10)$$\begin{array}{l} R_{3}\left(j\omega \right)=F\left(r_{3}\right)\\ =N_{v}\left(j\omega \right){\left(A_{0}\pi \left(e^{j\varphi_{0}}\delta \left(\omega +2\pi f_{0}\right)+e^{-j\varphi_{0}}\delta \left(\omega -2\pi f_{0}\right)\right)+\sum_{i=1}^{M}A_{ai}\pi \left(e^{j\varphi_{i}}\delta \left(\omega +2\pi f_{ai}\right)+e^{-j\varphi_{i}}\delta \left(\omega -2\pi f_{ai}\right)\right)\right)}^{*}\\ +N_{a}{\left(j\omega \right)}^{*}\left(A_{0}\pi \left(e^{j\varphi_{0}}\delta \left(\omega +2\pi f_{0}\right)+e^{-j\varphi_{0}}\delta \left(\omega -2\pi f_{0}\right)\right)+\sum_{i=1}^{N}A_{vi}\pi \left(e^{j\varphi_{i}}\delta \left(\omega +2\pi f_{vi}\right)+e^{-j\varphi_{i}}\delta \left(\omega -2\pi f_{vi}\right)\right)\right) \end{array}$$

According to Appendix A and characteristic of correlation [26,27], *R*_1_ >> *R*_2_ + *R*_3_, the frequency spectrum function of the correlation sequence is simplified as below:(11)$$\begin{array}{ll} R_{va}\left(j\omega \right) & =F\left(r_{va}\left(\tau \right)\right)\\ & \approx A_{0}^{2}\pi^{2}\left(\delta^{2}\left(\omega +2\pi f_{0}\right)+\delta^{2}\left(\omega -2\pi f_{0}\right)\right) \end{array}$$
where *r_va_*(*τ*) is the correlation sequence of *v*(*t*) and *a*(*t*), and *R_va_*(*j**ω*) is the Fourier transform of *r_va_*(*τ*). According to the characteristic of cross-correlation, the frequency spectrum of the *r_va_*(*τ*) contains only *f*_0_, which is the fundamental frequency of vibration and acoustic signals, and the interference frequencies *f_vi_* and *f_aj_* are both eliminated. As the fundamental frequency information remains in the correlation sequence of vibration and acoustic signals while interference and noise are restrained, the estimation of fundamental frequency can be achieved by applying frequency-domain methods to the correlation sequence.

### 2.3. Estimation of Fundamental Frequency 

In order to estimate the fundamental frequency of the correlation sequence obtained by the cross-correlation algorithm, a frequency-domain method is applied. The flowchart of the frequency estimation algorithm is shown in Figure 2. 

As shown in Figure 2, the discrete Fourier transform is first applied to the correlation sequence of vibration and acoustic signals. As the fundamental frequency component is enhanced while interference components are restrained by the cross-correlation algorithm, the spectral line with the largest amplitude represents the fundamental frequency component. The second step is to find the spectral line with the largest amplitude and four extra spectral lines surrounding it. Then, to improve the accuracy of the result, the energy centrobaric correction method [28] is applied. In this paper, five spectral lines are utilized. Assuming that *G_j_* is a peak value of the spectral line, and the subscript *j* indicates the index of the spectral line, *f_s_* is the sampling frequency and *L* is the number of samples, the estimated frequency *f* can be obtained as:(12)$$f=\frac{\sum_{j=-m}^{m}\left(j_{0}+j\right)G_{j_{0}+j}}{\sum_{j=-m}^{m}G_{j_{0}+j}}\frac{f_{s}}{L}\begin{array}{cc} , & m=2 \end{array}.$$

## 3. Hardware Design and Test Results

In order to evaluate the performance of the proposed method, an embedded hardware for rotation speed measurement was designed, and a test was carried out in an automobile inspection station. Considering the application scenario and reliability, the sensors were assembled in the form of a magnetic suction probe. Signals were transmitted through the coaxial line. Vibration and acoustic signals were sampled from the engine hood of an automobile, and a comparison of results was made between the proposed method and a traditional method based only on the vibration signal.

### 3.1. Hardware Design

A diagram of the hardware structure is shown in Figure 3, and a photo of the device is shown in Figure 4. According to the measurement principle, the vibration and acoustic signals related to the rotation speed were mainly distributed in the low-frequency range, so the frequency response of the sensor was set to the range from 0 to 250 Hz. In this paper, an MMA1220KEG micro-machined accelerometer was used as a vibration sensor. This chip features signal conditioning, a 4-pole low-pass filter, and temperature compensation. The acoustic sensor consisted of an electret microphone and a second-order Butterworth low-pass filter circuit.

An LPC1768 processor was deployed to implement the algorithm of rotation speed measurement. The LPC1768 operates at CPU frequencies of up to 100 MHz, which completely fulfilled the demands of the measurement task. 

### 3.2. Test Results

The experiment was carried out in an automobile inspection station, the device was tested under a practical situation, and the automobile was randomly chosen. The sensors were assembled in the form of a magnetic suction probe and they were attached on the engine hood of the tested automobile. For comparison, a traditional method for rotation speed measurement based on the vibration signal was also tested. All data was recorded with a data acquisition card for comparison. The test scenario and device are shown in Figure 5.

The sample rate was set to 512 Hz. The time frequency analysis results are presented in Figure 6. Figure 6a presents the results based on vibration signals and Figure 6b the results based on the cross-correlation of vibration and acoustic signals. To achieve a joint time-frequency analysis of vibration and acoustic signals synchronously, a sliding discrete Fourier transform and cross-correlation were combined. The window length was set to be 256 points with a 240-point overlap, and in each frame a discrete Fourier transform was applied on the correlation sequence of vibration and acoustic signals obtained by cross-correlation. The difference between Figure 6a and Figure 6b is evident. There was a difference between the frequencies of the vibration and acoustic signals, and the cross-correlation method was able to suppress the interference frequency components and enhanced the fundamental frequency, related to the rotation speed. This result indicates that cross-correlation analysis of acoustic and vibration signals provided a better performance in extracting the fundamental frequency component.

The results of rotation speed measurement based on the proposed method and the method solely based on the vibration signal are shown in Figure 7. It is obvious that the difference was mainly distributed in the idling stage and accelerating stage, and those two stages were the most disturbed stages in rotation speed measurement. Zoomed-in figures of boxes A and B in Figure 7 are shown in Figure 8a and Figure 8b, respectively. 

From Figure 8a, in the idling stage, the interference resulted in a large deviation in the vibration method. Meanwhile, the proposed method showed a more reliable performance in the idling stage. Figure 8b shows the measurement results during the accelerating stage. In this stage, the results of vibration method showed a large deviation, while the proposed method provided a correct and robust measurement result. The comparison shows that the proposed method had a better ability to attenuate interferences than the vibration method.

## 4. Conclusions

Noncontact measurements are of great significance to improve the efficiency of automobile inspection. Unfortunately, a robustness against external interference is a flaw of this method, which means that measurement results are no longer reliable in the presence of intense interference. This paper proposed a method based on cross-correlation to measure the rotation speed of an engine by utilizing the correlation of vibration and acoustic signals. By taking into consideration an acoustic signal and applying cross-correlation with a vibration signal, the interference signal was restrained and the fundamental frequency component, which is closely related to rotation speed, was enhanced. The proposed method could provide a reliable result with good anti-interference efficiency. Test results showed that the proposed method had a better performance than the vibration-based method in measuring the rotation speed of an engine. Embedded hardware was designed to implement the method, and the test results showed good reliability in an exemplary practical application. Further work will focus on how to realize the auto-recognition of the cylinder numbers, as the present work required prior knowledge of the cylinder numbers.

## Figures and Tables

**Figure 1 sensors-20-00683-f001:**
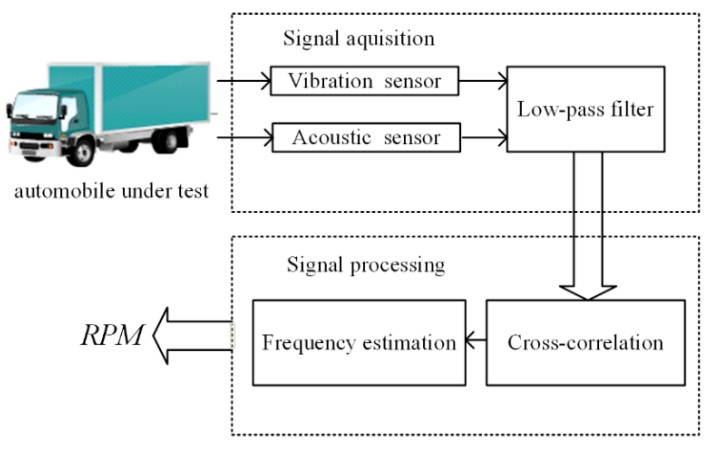
Principle of the measurement system. *RPM*: revolutions per minute.

**Figure 2 sensors-20-00683-f002:**
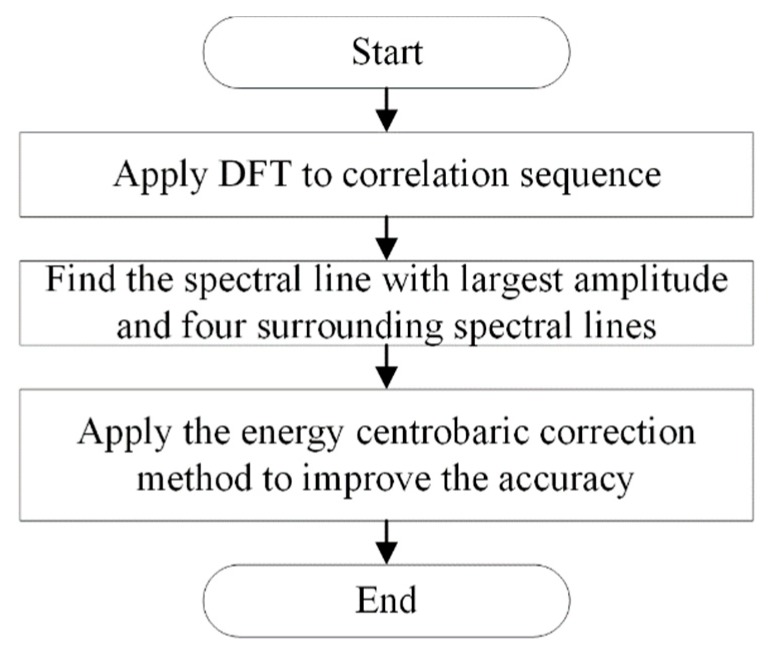
Flowchart of *RPM* measurement. DFT: discrete Fourier transform.

**Figure 3 sensors-20-00683-f003:**
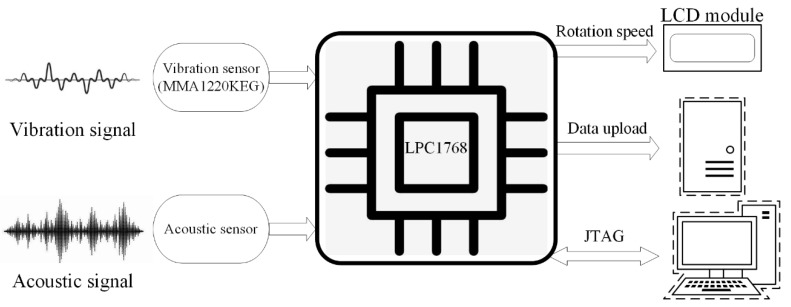
Diagram of the hardware structure. JTAG: Joint Test Action Group

**Figure 4 sensors-20-00683-f004:**
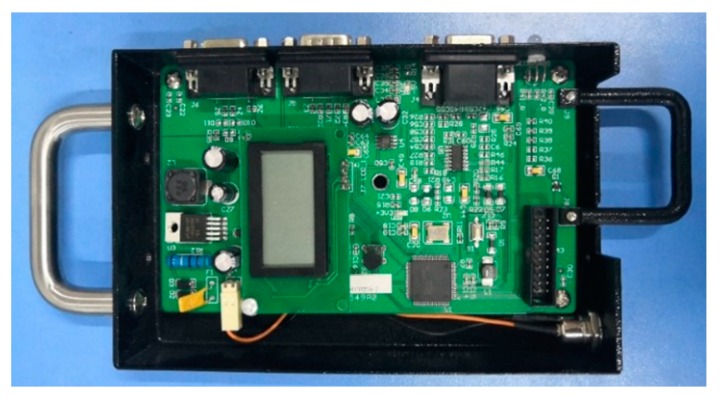
Photo of the device.

**Figure 5 sensors-20-00683-f005:**
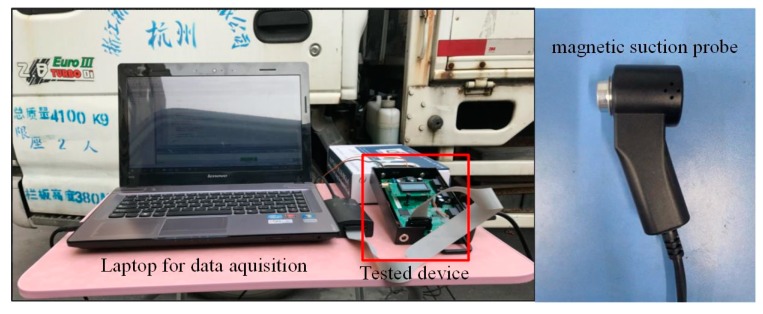
Photo of test scenario and device.

**Figure 6 sensors-20-00683-f006:**
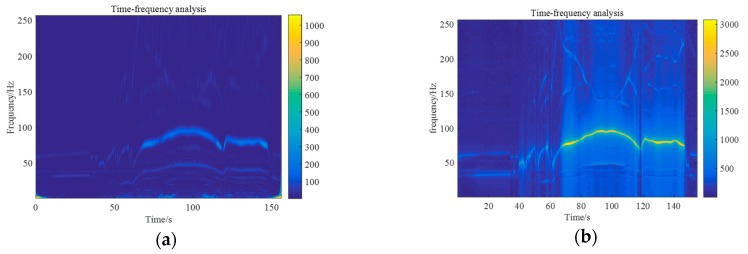
(**a**) Time-frequency analysis of the vibration signal; (**b**) Time-frequency analysis of vibration and acoustic signals.

**Figure 7 sensors-20-00683-f007:**
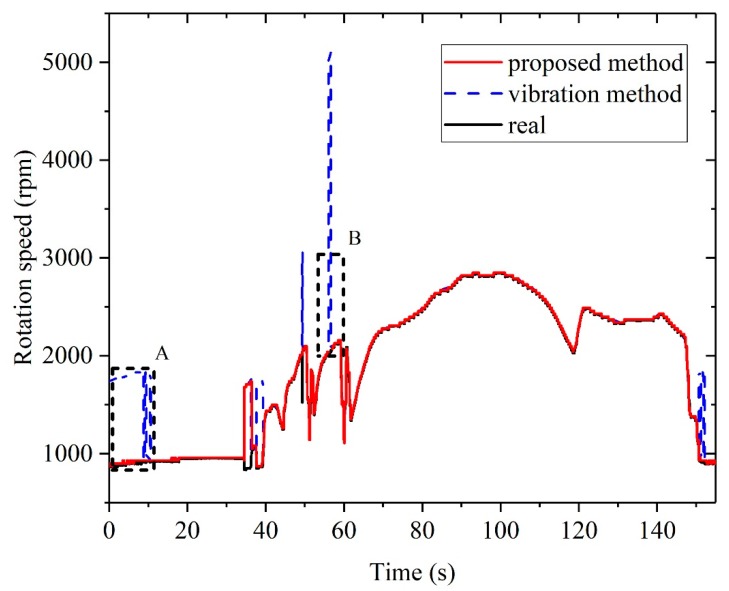
Test results of rotation speed measurement.

**Figure 8 sensors-20-00683-f008:**
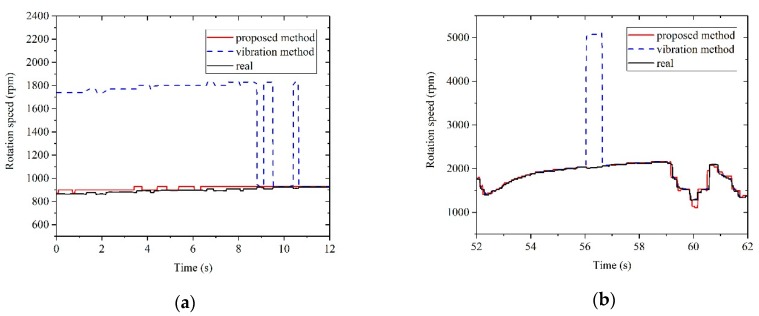
(**a**) Zoomed-in figure of box A in Figure 7; (**b**) Zoomed-in figure of box B in Figure 7.

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
