# Peer review of "Analysis of Vibration and Acoustic Signals for Noncontact Measurement of Engine Rotation Speed"

_sensors, 2020, doi:10.3390/s20030683_

Round 1
Reviewer 1 Report
The paper deals with a procedure to compute the engine rotation speed by means of non contact measurements. The paper is well organized and the subject is adequately presented.
The paper is worth to be published after the following minor revision will be accomplished:
Equations 3 and 4, please explain the meaning of "M" and "N". Is "N" the same of equation 12? If not, please use a dyfferent letter to avoid misunderstandments; Equation 5: add "dt" at the end of the integral; Please specify the integration interval in equations 5, 6 and 7; in equations 8, 9 and 10 the letter "f" is used to indicate the Fourier transform of a function while in equation 11 the letter "F" is used apparently for the same transform. Please correct the notation; in equations 8 to 11, the function "δ" (\delta) is introduced, please specify what function stands for; in equations 8 to 11, the symbol "φ" (\phi) appears but it is not explained, please explain; with respect to equations 8 to 11, they report the fourier transforms of the cross correlation functions "r_j". It is well known that the fourier transform is defined as a convolution integral of the function to be transformed by an exponential function kernel in continuos time domain or, if you consider discrete time domain, the discrete fourier transform (DFT) is defined by a summation of N terms (where N is the number of samples). Please specify how those equations have been obtained (maybe by adding an Appendix section where all the steps can be reported without compromising the readibility of the paper); The sentence from line 107 to 109 ("According ... ...eliminated.") need to be supported by formulas and/or references; Lines 157 to 164 and Figure 5: explain better how you build time frequency diagrams by using the proposed algorithm. For instance, how many samples are used for the cross-correlation, how many time per second the analysis is done and so on; Lines 167 to 178 and Figures 6 and 7: it is suggested to compare the proposed and vibration methods to the real value of the vibration (acquired by a well established literature method) otherwise one cannot know a priori what is the correct value.Author Response
Thank you for your comments, please see the attachment.

Reviewer 2 Report
The article deals with the problem of measurement of engine rotate speed. Authors propose to use the cross-correlation between vibration and acoustic signals. It is novel and very interesting approach. The structure of the manuscript is correct and the method is well explained. Results provided in the manuscript indicate that proposed method is better than the traditional vibration-based measurement method. Therefore I think that this work is worth to publish. However, I have some comments and questions to authors.
I think that in the introduction authors should give more information about correlation methods that are applied in various field of mechanical engineering. These are for example Pearson and Spearman correlation coefficients. Apart from that the cross-correlation function has been applied to compare surfaces of rotary elements, which was described for example in work Quantitative comparison of cylindricity profiles measured with different methods using Legendre-Fourier coefficients, Metrology and Measurement Systems, XVII (2010), No. 3, pp. 397−404 by Adamczak et al. I am very interested in the software that was used to detremine the correlation. Could authors give some more information about it? Was it developed in Matlab? I suppose that it would be very good if authors can give source codes of the procedures developed (of course, if it is possible).Generally, I think that the manuscript is of high scientific quality and it can be interesting for the readers of the journal.
Author Response
Thank you for your comments, please see the attachment.

Reviewer 3 Report
The problems I have detected are related to figures, as described below. It seems to me a good work, written correctly and, above all, clearly.
The method shown is interesting as far as there are limits, which the authors have clearly highlighted.
Figure 2 is repeated twice (with different images) and this entails an error in the numbering of the subsequent images. I suggest solving the problem by eliminating Fig. 1 which appears superfluous with respect to the content of the paper. The equation 2 to which it refers and the subsequent description are more than sufficient.
Fig. 6: the results shown are not so obvious as stated in the paper, I suggest to use a blue dotted line to allow the reader to follow up also the red line behavior, as it is in figs. 7.
Author Response

(The authors gave the same response as above.)
